# PICK-OR-MIX: DYNAMIC CHANNEL SAMPLING FOR CONVNETS

## ABSTRACT

Channel squeezing is a crucial operation in convolutional neural networks (ConvNets). It is carried out via $1 \times 1$ convolution layers and dominates a large portion of computations and parameters of a given network. ResNet-50, for instance, consists of 16 such layers, forming 33% of total layers and 25% (1.05B/4.12B) of total FLOPs. In light of their predominance, we present a new multi-purpose module for dynamic channel sampling, namely Pick-or-Mix (PiX). PiX divides a set of channels into subsets and then picks from them, where the picking decision is dynamically made per each pixel based on the input activations. We show that PiX allows ConvNets to learn better data representation than vanilla channel squeezing in far fewer computations. We plug PiX into prominent ConvNet architectures and verify its multi-purpose utilities. After replacing $1 \times 1$ channel squeezing layers in the ResNet family with PiX, the networks become 25% faster without losing accuracy. We also show that PiX can achieve state-of-the-art performance on network downscaling and dynamic channel pruning.

**Code:** Will be released post reviews. **Supplement:** Please see Appendix.

## 1 INTRODUCTION

Convolutional neural networks (ConvNets) (Simonyan & Zisserman, 2014; He et al., 2016) have been successfully applied to many machine vision tasks (Ren et al., 2015; Jeong et al., 2022). With the introduction of larger models, a general trend is to make them faster via channel pruning. Prior works in channel pruning (He et al., 2018a; Hua et al., 2019; Gao et al., 2018; Han et al., 2020) focus on making network lighter to accelerate the inference speed. However, some approaches require specialized convolution implementations and pre-trained models (Gao et al., 2018), or they are constrained by the baseline accuracy (Han et al., 2020). Moreover, whether static or dynamic, these channel pruning methods permanently remove or deactivate the network channels, thus hindering the network from handling difficult inputs (Gao et al., 2018; Tang et al., 2021).

It is a fundamental property of ConvNets that for a given spatial location or pixel in the ConvNet feature map, any one channel may have stronger activation, thus of considerable importance, while for another pixel, the same channel might be less important. Therefore, it is crucial to allow the network to *prioritize channels differently per each pixel* instead of dropping a whole channel applied by pruning approaches. This inspires us to pick neuron-specific output from the channels instead of shutting down an entire channel. In addition, we observe that standard ConvNet designs still have room for improvement, i.e., $1 \times 1$ convolution layers (or called channel squeezing layers) dominate in both number and computations without contributing to the receptive field due to their pixel-wise operation nature. For instance, ResNet-50 (He et al., 2016) consists of 16 such layers out of 50, accounting for $\sim 25\%$ (1.05B/4.12B) of overall FLOPs.

In this context, we introduce a novel module, namely Pick-or-Mix (PiX) that addresses the computational dominance of channel-squeezing layers by *dynamically sampling channels*, PiX transforms a feature map $X \in \mathbb{R}^{C \times H \times W}$ into another one $Y \in \mathbb{R}^{\lceil C/\varsigma \rceil \times H \times W}$. Essentially, our method picks or mixes $\lceil C/\varsigma \rceil$ channels from the input $C$ channels with a sampling factor $\varsigma$. It divides a set of channels into subsets and then outputs one channel from each subset via our Pick-or-Mix strategy. PiX samples channel based on the *pixel-level runtime decisions* made by the preceding layers; thus, decisions of PiX are dynamic and input-dependent. In addition, Pick-or-Mix does not involve extensive pixel-wise convolution, making the network more efficient. The simple design allows us to plug PiX into representative ConvNets. We plug PiX into representative ConvNets for the purpose of faster channel squeezing, network downscaling, and dynamic channel pruning.

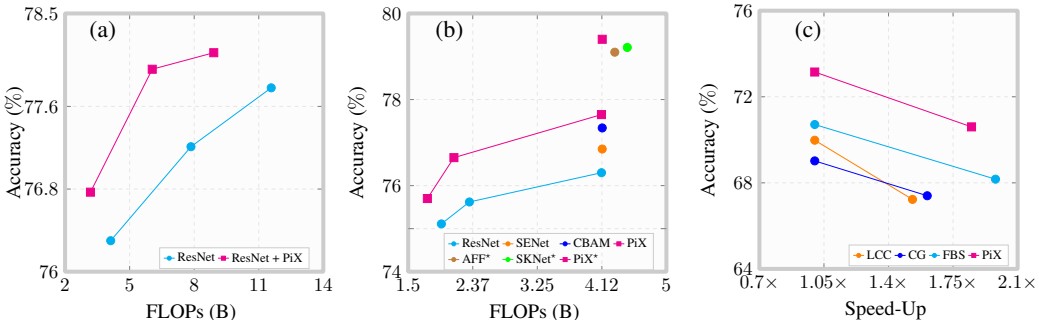

Figure 1: FLOPs and Top-1 accuracy of PiX operating in different modes. (a) PiX in fast channel squeezing modes turns ResNet-50, 101, and 152 (from left to right) faster by 25% fewer FLOPs (Table 2). (b) network downscaling on ResNet-50 (beginning from 4.51B FLOPs) with different downscaling factors. A '⋆' denotes 160 epochs. PiX downscaled ResNet-50 outperforms existing approaches that aim to improve accuracy via feature fusion or other techniques (Table 6). (c) dynamic channel pruning on ResNet-18. Our method outperforms recent approaches by large margin (Table 4).

Our experiments show that PiX can reduce the computational cost of the vanilla channel squeezing layer while maintaining or achieving even better performance, e.g., ResNet becomes $\sim 25\%$ faster without bells and whistles (Sec 3.4.1, Table 2). PiX can customize ConvNets in a controlled manner while being faster and more accurate than the baseline counterpart with similar parameters (Sec. 3.4.2, Table 3), e.g., PiX outperforms recent RepVGG (Ding et al., 2021) without a complicated training phase while having simple network design. We also observe similar accuracy but at reduced parameters (Table 6). PiX performs better by $\sim 3\%$ relative to various recent dynamic channel pruning approaches (Gao et al., 2018; Chen et al., 2020; Tang et al., 2021; Park et al., 2023) on ResNet18 with $\sim 2\times$ FLOPs saving. (Sec. 3.4.3, Table 4).

We show the accuracy and FLOPs of PiX with other state-of-the-art approaches in Figure 1. We also conducted transfer learning on PiX-enhanced network on CIFAR-10, CIFAR-100 for classification, and CityScapes for semantic segmentation. We observe better performance relative to the baselines.

## 2 RELATED WORK

**Convolutional Neural Networks**   The earlier ConvNets (Simonyan & Zisserman, 2014; He et al., 2016) are accuracy-oriented but still dominant in the industry (Kumar & Behera, 2019; Ding et al., 2021), thanks to their high representation power, architectural simplicity, and customizability. EfficientNet (Tan & Le, 2019) emerged with network architecture search, but due to its nature of AutoML, it is deep and branched compared to traditional ConvNets (Simonyan & Zisserman, 2014; He et al., 2016). Even after half a decade, ResNet continues to improve (Dai et al., 2021; Li et al., 2019), indicating its architectural significance, while VGG-like architecture continues as it is design-friendly with low-powered computing devices due to its shallow, easily scalable, and low latency design (Kumar et al., 2020).

This is also visible from ResNet design space exploration (Radosavovic et al., 2020) that provides a competitive alternative to the advanced ConvNets (Tan & Le, 2019) while being simpler. SENet (Hu et al., 2018), CBAM (Woo et al., 2018), and ResNest (Zhang et al., 2022), Attentional Feature Fusion (Dai et al., 2021) further depict the importance of older architectures by developing novel units to improve the accuracy of ResNet by adding parameters and marginal computational overhead. More recently, RepVGG (Ding et al., 2021) improves the inference of years old VGG (Simonyan & Zisserman, 2014) model. Therefore, there are some design choices of ConvNets left untouched, with room for improvement. In this paper, we tackle the challenges remaining in squeezing layers.

**Accelerated Inference**   ConvNet acceleration begins with static pruning (Li et al., 2016) or network compression (He et al., 2018b). These methods (Li et al., 2016; He et al., 2018b) are model agnostic, but they require the additional overhead of pre-training and fine-tuning, thus increasing the training time (Gao et al., 2018). Whereas, our PiX is free of such issues and offers a promising alternative to these approaches. Furthermore, by using more efficient convolutions such as depthwise separable convolution (Sifre & Mallat), MobileNets (Howard et al., 2017; Sandler et al., 2018; Zhang et al.,

2018) address this issue at the network architecture level. In the same manner, our PiX is also an architectural enhancement that can function in multiple ways as mentioned earlier.

## 3 PICK-OR-MIX (PIX)

Modern ConvNets (He et al., 2016; Ding et al., 2021; Zhang et al., 2022) are essentially a stack of convolution layers, but the design of channel squeezing still has room for improvement. The main challenge is appropriately exploiting the cross-channel information and developing a suitable mixing strategy to ensure accurate model learning. In this work, we introduce Pick-or-Mix (PiX).

**Overview**  Consider a tensor $X = \{X^{[1]}, X^{[2]}, X^{[3]}, ..., X^{[C]}\}$, where $X^{[i]} \in \mathbb{R}^{H \times W}$ denotes $i^{th}$ channel of $X$. We aim to produce $Y = \{Y^{[1]}, Y^{[2]}, Y^{[3]}, ..., Y^{[\lceil C/\zeta \rceil]}\}$ where $\zeta$ is the sampling factor, such that $O(\mathcal{F}_{pix}) \ll O(\mathcal{F}_s)$, where $\mathcal{F}_{pix}$ is the PiX enhanced network and $\mathcal{F}_s$ is the original network. To achieve that, the proposed dynamic channel sampling approach (PiX) progressively infers intermediate 1D descriptors $z \in \mathbb{R}^C$, $p \in \mathbb{R}^{\lceil C/\zeta \rceil}$ from input feature map $X \in \mathbb{R}^{C \times H \times W}$ for channel sampling by using learnable parameter $\phi = \{\theta, \beta\}$, and applies per-pixel dynamic channel sampling operator $\pi$ by fusing subset of channels. An output feature map $Y \in \mathbb{R}^{\lceil C/\zeta \rceil \times H \times W}$ of reduced dimensionality is controllable by a sampling factor $\zeta \in \mathbb{R}_{\geq 1}$.

The PiX module is illustrated in Figure 2 and can be sectioned into three stages: (1) global context aggregation, which provides a channel-wise global spatial context in the form of $z$ (Sec. 3.1) (2) cross-channel information blending that transforms $z$ into $p$, referred to as PiX sampling probability (Sec. 3.2), and (3) channel sampling stage that utilizes $p$ and $X$ to produce $Y$. (Sec. 3.3)

### 3.1 GLOBAL CONTEXT AGGREGATION

We define a transformation of global context aggregation as $gca : \mathbb{R}^{C \times H \times W} \to \mathbb{R}^C$ which gathers global spatial context from the input $X$ for each channel:

$$gca(X) = \frac{1}{H \times W}\left[ cc\left(X^{[0]}\right), cc\left(X^{[1]}\right), ..., cc\left(X^{[C-1]}\right) \right] \tag{1}$$

where, cc $: \mathbb{R}^{H \times W} \to \mathbb{R}$ reduces $i^{th}$ channel $X^{[i]}$ of $X$ to a scalar. We use $l_1$-norm for cc due to its computational efficiency and vectorized parallelization onto GPUs. $l_1$-norm of a channel is also known as global pooling, which is commonly employed (He et al., 2016; Hu et al., 2018) to aggregate global spatial information in a computationally efficient manner.

### 3.2 SAMPLING PROBABILITY

Now the output of the previous step $z = gca(X)$ is passed through *sampling probability predictor* $\phi$, serving two purposes. First, since each element of $z$ consists of spatial information of only a single channel of $X$, the descriptor $z$ lacks cross-channel information. $\phi$ mitigates this issue by blending the cross-channel information in the elements of $z$. Second, the fusion factor $\zeta$, i.e., $C$ to $\lceil C/\zeta \rceil$, reduces the input number of channels. We define $\phi(z) = z\theta + \beta$, where, $\theta \in \mathbb{R}^{\lceil C/\zeta \rceil \times C}$ and $\beta \in \mathbb{R}^{\lceil C/\zeta \rceil}$ are the weights and the biases, initialized with *Xavier* (Glorot & Bengio, 2010) and zero respectively. After $\phi(z)$, we obtain channel sampling probability $p$ with sigmoid function, $p \in \mathbb{R}_{\geq 0}^{\lceil C/\zeta \rceil}$ which is used in Sec. 3.3 to optimize channel sampling for richer spatial and channel context.

### 3.3 DYNAMIC CHANNEL SAMPLING

This section describes our computationally efficient dynamic channel sampling approach conditioned on $p$. For that, we define a functor $\mathcal{F} : \mathbb{R}^{C \times H \times W} \to \mathbb{R}^{\lceil C/\zeta \rceil \times H \times W}$ such that $Y = \mathcal{F}(X; p)$.

**Channel Space Partition**  The foremost step of channel sampling is partitioning $X$ into $\lceil C/\zeta \rceil$ subsets. Each subset ($\Gamma^{[i]}$, where $i \in \{0, \cdots, \zeta - 1\}$) receives a maximum of $\zeta$ channels with the last one lesser than that in case $C/\zeta$ is non-integer.

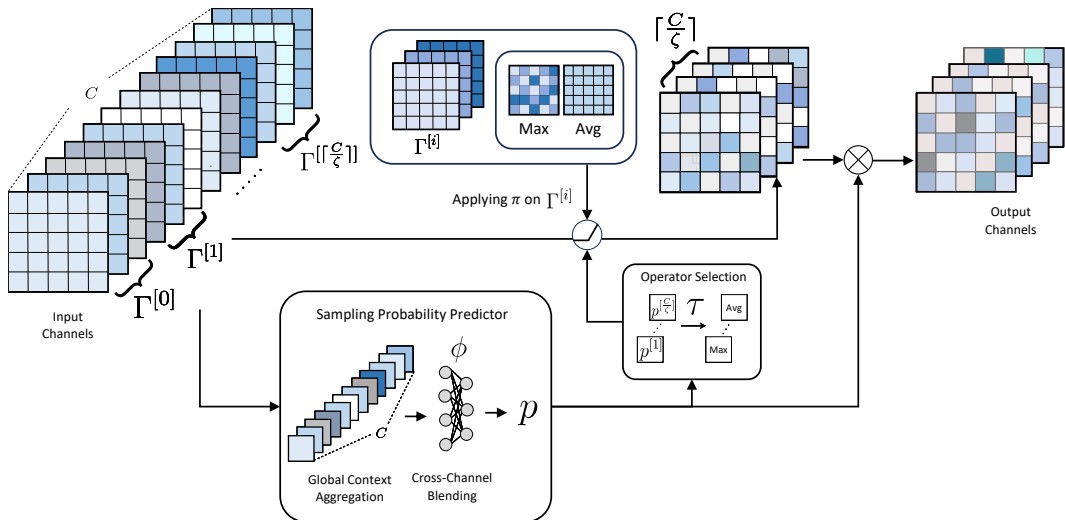

Figure 2: The proposed PiX module with its Pick-or-Mix dynamic channel sampling strategy (PiX).

**Per-pixel Channel Fusion**  We devise a channel fusion strategy, namely *Pick-or-Mix* for each partitioned channel subset $\Gamma^{[i]}$. Specifically, for any spatial location (or we denote *pixel*, for the sake of simplicity) on $\Gamma^{[i]}$, we consider a vector $v \in \mathbb{R}^\zeta$. We then apply the channel fusion strategies to obtain a single scalar that defines the value of the output channel. $v$ is fused via the following equations:

$$\pi(v) = \begin{cases} p^{[i]} \times \texttt{Max}(v), & p^{[i]} \leq \tau \\ p^{[i]} \times \texttt{Avg}(v), & p^{[i]} > \tau, \end{cases} \qquad (2)$$

where $\pi$ is *Pick* (selecting the maximum) or *Mix* (averaging responses) function that reduces $v$ to a scalar, $p^{[i]}$ is the sampling probability for a $i$-th subset (Sec. 3.2). $\tau$ is hyperparameter, set to $0.5$ based on our ablations. In Eq. 2, the selection of a fusion operator is performed dynamically via the sampling probability $p$ produced via the input, thus making PiX input adaptive.

It is important to note that Eq. 2 is applied on a *per-pixel basis*, indicating that *channel sampling is applied differently for each pixel*. Specifically, when $\zeta > 1$, with the help of $\texttt{Max}$, the selected channel index in a subset may differ for each pixel, although the sampling probability $p^{[i]}$ for each pixel in the subset is the same. In addition, each pixel spans all the $\Gamma^{[i]}$ subsets, and each subset may apply a different operator, i.e., some subgroup applies $\texttt{Max}(\cdot)$, and the other applies $\texttt{Avg}(\cdot)$. This is why PiX allows the network to *select channels on a per-pixel basis*, bringing more flexibility. Note that $\pi$ just refers to pre-computed $p^{[i]}$ for the decision, *not involving pixel-wise $1 \times 1$ convolutions*. This scheme can save computation costs. When $\zeta = 1$, Eq. 2 scale the channels by $p$ (since $\texttt{Max}(v) = \texttt{Avg}(v)$), and it will act as global channel-wise attention as in SENet (Hu et al., 2018).

To generalize over the whole input feature map $X$, the functor $\mathcal{F}$ for this strategy can be given as:

$$\mathcal{F}(X;p) = \left[\pi\left(\Gamma^{[0]}\right), \ \pi\left(\Gamma^{[1]}\right), \ \pi\left(\Gamma^{[2]}\right), \ ..., \ \pi\left(\Gamma^{[\lceil C/\zeta \rceil - 1]}\right)\right] \qquad (3)$$

The PiX process is visualized in Figure 2.

Our motivation to selectively utilize $\texttt{Max}$ and $\texttt{Avg}$ lies in the fundamentals of ConvNets (Krizhevsky et al., 2012) where max and average pooling performs essentially a summarization operation. The dynamic decision based on $p^{[i]}$ enables the ConvNets to learn rich representations and allows sub-sampling of the features. We also support our motivation empirically by employing the *minimum* operator instead of $\texttt{Max}$ or $\texttt{Avg}$. We observe a performance degradation by roughly $2\%$ (See Table A5 and Sec. M in Appendix). Since fusion is done *on a pixel basis*, one pixel may prioritize any channel over another, while any other pixel may suppress that channel by prioritizing another, demonstrating the capability of PiX to fuse the channels at the pixel level. This degree of freedom to fuse values dynamically introduces a high level of non-linearity into the network, which helps to achieve PiX a competitive accuracy on various tasks (discussed next) with a simplified network structure.

### 3.4 PiX Embodiment as a Multi-Purpose Module

The ability of PiX to perform channel sampling naturally translates to the underlying operations of different tasks, such as channel squeezing (Sec. 3.4.1), network scaling (Sec. 3.4.2), and dynamic channel pruning (Sec. 3.4.3). We describe below in detail how PiX achieves these objectives despite keeping its structure the same. We also discuss the benefit of using PiX for these tasks.

#### 3.4.1 Fast Channel Squeezing

Prior works have conducted channel squeezing operations mostly with $1 \times 1$ layers in ResNet-like designs (He et al., 2016). PiX maintains a similar level of accuracy to such approaches by utilizing channel sampling probability (Sec. 3.2) in conjunction with the pixel-wise dynamic channel sampling (Sec. 3.3). More importantly, PiX is free from expensive dense $1 \times 1$ convolution. Instead, by operating on a vector $z$, PiX effectively saves FLOPs and squeezes the channel faster.

To demonstrate our claims, we *replace channel squeezing* $1 \times 1$ *layers* in the representative ResNet (He et al., 2016) family (ResNet-50, -101, and -152) with PiX and evaluate the accuracy, FLOPs, and training and inference time. PiX speeds up the training and inference, which are empirically verified in Table 2 and Table A2. See Figure A1a in the Appendix for the details. To check our claim's possibility on the network other than ConvNets, we also conducted an experiment on the *state-of-the-art* Efficient ViT (Liu et al., 2023), and the following is discussed in Sec. D Appendix. Interestingly, *EfficientViT Transformer becomes* 27% *faster at* 0.68% *with better accuracy*.

Alternatively, channel squeezing can be done via depth-wise pooling in a non-parametric way (Hussain & Hesheng, 2019). However, it eliminates all the squeeze convolution layers, resulting in an accuracy drop (see E4, Table 6).

#### 3.4.2 Network Downscaling

We can control ConvNets' parameters and computational complexity by adjusting the number of input or output channels. When conducting parameter reduction, it is called network downscaling. PiX can achieve this goal via its channel reduction capability. In our approach, the input feature map for each layer is squeezed by the PiX module with sampling factor $\zeta > 1$ and then sent to the next convolution layer.

PiX can downscale ConvNets by inserting PiX modules into the existing layers, allowing it to control the network in downscaling by changing $\zeta$. We use ResNet-18, ResNet-50, and VGG-16 for the effectiveness of this application. Please refer to Figure A1b, A1c, A1d in Appendix. Interestingly, PiX-downscaled network variant consistently outperforms the downscaled baseline. PiX-downscaled networks have the same parameters but lower FLOPs and higher accuracy (Table 3).

#### 3.4.3 Dynamic Channel Pruning

When we plug PiX into a model, it uses $\zeta$ to determine the number of output channels. Thus, once $\zeta$ is set, the number of channels obtained from PiX is deterministic or *static*. However, as PiX selects channels on the fly, meaning that which channels will be sent to the next layer is not predetermined, it leads to a *dynamic* reduction behavior.

For this reason, we call PiX as static-dynamic channel pruner. This contrasts with the dynamic channel pruning approach, which keeps all the channels in the network intact but decides which ones to compute to save computations. This mandates the need for *specialized convolution implementation* to take advantage. On the other hand, the static-dynamic behavior of PiX is *free of such necessity*, which is of practical significance. The static behavior reduces the network's memory footprint and bandwidth while outperforming dynamic channel pruning approaches.

See Figure A1c, A1d in Appendix for the procedure to embody PiX as dynamic channel pruner. Table 4 and 5 show a comparison with dynamic pruning approaches. We use ResNet-18 and VGG-16 for evaluation.

### 3.5 Relation With Existing Approaches

We discuss representative approaches that are closest to the proposed PiX. The idea of using global context was introduced by SENet (Hu et al., 2018) aiming to improve network accuracy, which squeezes and expands a global context vector by using two convolution layers to predict channel

Table 1: A functional comparison of PiX.

| Method | No Finetuing | No Custom Convolutions | As a Channel Squeezer | As a Network Downscalar | As a Dynamic Pruner |
|---|---|---|---|---|---|
| ● SE Hu et al. (2018) | ✓ | ✓ | ✗ | ✗ | ✗ |
| ● CBAM Woo et al. (2018) | ✓ | ✓ | ✗ | ✗ | ✗ |
| ● FBS Gao et al. (2018) | ✗ | ✗ | ✗ | ✗ | ✓ |
| ● PiX | ✓ | ✓ | ✓ | ✓ | ✓ |

Table 2: PiX as a fast channel squeezer. We replace $1 \times 1$ channel squeezing layers in ResNet with PiX. We denote the channel squeezing factor of the vanilla network and our modification in the $\zeta$ column.

| | Architecture | $\zeta$ | #Param | FLOPs ↓ | Top-1% ↑ | Train Time Per-Iteration ↓ | Train Time 120-Epochs ↓ | Train Time 200-Epochs ↓ |
|---|---|---|---|---|---|---|---|---|
| E0 | ● ResNet-50 He et al. (2016) | 4 | 25.5M | 4.12B | 76.30 | 575ms | 4.0 Days | 6.7 Days |
| | ● ResNet-50+PiX | 4 | 25.5M | **3.18B** (↓22.8%) | **76.77** (↑0.47%) | **359ms** | **2.5 Days** | **4.1 Days** |
| | ● ResNet-50+PiX@Avg | 4 | 25.5M | **3.18B** (↓22.8%) | **76.58** (↑0.28%) | **359ms** | **2.5 Days** | **4.1 Days** |
| | ● ResNet-50+PiX@Max | 4 | 25.5M | **3.18B** (↓22.8%) | **76.57** (↑0.27%) | **359ms** | **2.5 Days** | **4.1 Days** |
| E1 | ● ResNet-101 He et al. (2016) | 4 | 44.5M | 7.85B | 77.21 | 575ms | 4.0 Days | 6.7 Days |
| | ● ResNet-101+PiX | 4 | 44.5M | **6.05B** (↓22.9%) | **77.96** (↑0.45%) | **431ms** | **3.0 Days** | **5.0 Days** |
| E2 | ● ResNet-152 He et al. (2016) | 4 | 60.1M | 11.58B | 77.78 | 863ms | 6.0 Days | 10.0 Days |
| | ● ResNet-152+PiX | 4 | 60.1M | **8.91B** (↓23.0%) | **78.12** (↑0.44%) | **575ms** | **4.0 Days** | **6.7 Days** |
| E3 | ● ResNet-50 He et al. (2016) | 8 | 12.3M | 1.85B | 73.66 | 260ms | 1.8 Days | 3.0 Days |
| | ● ResNet-50+PiX | 8 | 12.3M | **1.39B** (↓24.8%) | **74.47** (↑0.81%) | **180ms** | **1.25 Days** | **2.0 Days** |
| E4 | ● ResNet-50+SE Hu et al. (2018) | 4 | 28.0M | 4.13B | 76.85 | 575ms | 4.0 Days | 6.7 Days |
| | ● ResNet-50+SE Hu et al. (2018)+PiX | 4 | 28.0M | **3.19B** (↓22.8%) | **76.95** (↑0.10%) | **359ms** | **2.5 Days** | **4.1 Days** |

saliency. CBAM (Woo et al., 2018) extends SENet, performing both max-and avg pooling during global context extraction, then passes them through a shared MLP. FBS (Gao et al., 2018) uses global attention to predict channel saliency. FBS picks Top-K channels using the predicted channel saliency, and the suppressed channels are inhibited in the computations of the subsequent layer.

The proposed PiX differs from existing channel pruning (Gao et al., 2018) or channel squeezing (Hu et al., 2018) approaches. Our approach differs in structure and functionality. PiX *does not require an architectural change* to perform pixel-wise channel sampling. On the other hand, FBS (Gao et al., 2018), for instance, is a channel pruner, and the design is not intended for a channel squeezer. For reference, we report the accuracy drop when FBS is modified to work as a channel squeezer in Sec. 4.3. In addition, PiX is generally *applicable for multiple purposes by plugging it into existing networks*. Please refer to Sec. 3.4 and Figure A1a, A1b, A1c, and A1d to see how PiX is adopted for different purposes.

A functional comparison of PiX with previous approaches is shown in Table 1. We recommend seeing Figure A2 in the Appendix for visual differences between PiX and SENet, CBAM, and FBS. We also provide details on *the memory and FLOPs requirements* of PiX, SE, CBAM, and FBS in Sec. G in Appendix. Note that PiX has the lowest FLOPs and memory consumption.

## 4 EXPERIMENTS

We evaluate PiX on ImageNet (Deng et al., 2009) with 1.28M training and 50K validation images over 1000 categories. For transfer learning, we use CIFAR-10 and CIFAR-100 datasets for image classification and CityScapes (Cordts et al., 2016) for the downstream task of semantic segmentation. Please see Appendix Sec. L and K for training details and code snippets. Ablations are provided in Sec. M in Appendix. We use (flops counting tool) for FLOP calculations which aligns with our theoretical calculations as mentioned in Sec. F in the Appendix.

### 4.1 PIX AS FAST CHANNEL SQUEEZER (SEC. 3.4.1)

Fast Channel Squeezing aims to reduce FLOPs while maintaining accuracy and parameters. Refer to Table 2 for the evaluation.

**E0-E2.** PiX achieves computationally efficient squeezing, as visible by the $\sim 23\%$ reduction in FLOPs in all of the PiX variants. Interestingly, ResNet-101+PiX surpasses the baseline ResNet-152 with a significant FLOP difference of $47\%$. We argue that our conjecture on reusing the parameters of PiX works to maintain the non-linearity of the network is verified. Also, the empirical result shows

Table 3: PiX as Network DownScaler.

| Approach | #Param | FLOPs↓ | Top-1%↑ | Approach | #Param | FLOPs↓ | Top-1%↑ |
|---|---|---|---|---|---|---|---|
| ● ResNet-18 × 1.050 | 12.80M | 1.99B | 71.71 | ● VGG-16 × 1.05 | 16.72M | 4.20B | 73.25 |
| ● ResNet-18 + PiX @$\zeta = 1$ | 12.80M | **1.84B** | **73.15** | ● VGG-16 + PiX @$\zeta = 1$ | 16.78M | **3.85B** | **74.53** |
| ● ResNet-18 × 0.756 | 6.77M | 1.12B | 69.37 | ● VGG-16 × 0.63 | 8.67M | 2.26B | 70.53 |
| ● ResNet-18 + PiX @$\zeta = 2$ | 6.77M | **0.99B** | **70.60** | ● VGG-16 + PiX @$\zeta = 2$ | 8.65M | **1.94B** | **72.47** |
| ● ResNet-18 × 0.631 | 4.78M | 0.82B | 67.55 | ● VGG-16 × 0.75 | 5.97M | 1.59B | 69.12 |
| ● ResNet-18 + PiX @$\zeta = 3$ | 4.77M | **0.72B** | **68.70** | ● VGG-16 + PiX @$\zeta = 3$ | 5.96M | **1.32B** | **70.78** |
| ● ResNet-18 × 0.555 | 3.74M | 0.67B | 66.10 | ● VGG-16 × 0.54 | 4.59M | 1.25B | 67.56 |
| ● ResNet-18 + PiX @$\zeta = 4$ | 3.74M | **0.57B** | **67.15** | ● VGG-16 + PiX @$\zeta = 4$ | 4.59M | **0.98B** | **69.32** |
| ● ResNet-50 × 1.051 | 28.09M | 4.51B | 76.57 | ● MobileNet-v1 ×1.334 | 7.04M | 0.97B | 74.49 |
| ● ResNet-50 + PiX @$\zeta = 1$ | 28.08M | **4.13B** | **77.65** | ● MobileNet-v1 + PiX @$\zeta = 1$ | 7.03M | **0.58B** | **74.53** |
| ● ResNet-50 × 0.732 | 14.09M | 2.33B | 75.62 | ● MobileNet-v1 ×1.0 | 4.20M | 0.58B | 70.60 |
| ● ResNet-50 + PiX @$\zeta = 2$ | 14.08M | **2.12B** | **76.65** | ● MobileNet-v1 + PiX @$\zeta = 2$ | 4.06M | **0.33B** | **72.27** |
| ● ResNet-50 × 0.657 | 11.52M | 1.95B | 75.11 | | | | |
| ● ResNet-50 + PiX @$\zeta = 3$ | 11.51M | **1.76B** | **75.70** | | | | |

that PiX learns useful data representations. (Sec. 3.4.1), since despite the reduction in FLOPs, PiX exhibited slight accuracy improvements.

**E3.** We analyze PiX for a higher squeezing factor, i.e., $\zeta = 8$, and observe that PiX performs better than the baseline while having almost 25% fewer FLOPs. Interestingly, the accuracy gap between ResNet@$\zeta = 4$ and $\zeta = 8$ is 2.64%, while this gap reduces to 2.30% for PiX at a notable 56% reduction in the FLOPs. These empirical results demonstrate the robustness of PiX towards parameter reduction and its ability to learn to sample channels efficiently.

**E4.** We also test PiX in conjunction with SE Modules (Hu et al., 2018). It is noticeable that PiX performs better than the baseline, especially in FLOPs, indicating that PiX improves the computational performance of SE-like modules.

**Training Throughput.** Table 2 also shows throughput analysis on $8\times$ NVIDIA 1080Ti system. Noticeably, PiX has the lowest per-iteration time, which reduces the overall training duration, indicating that $1\times1$ squeeze layers could be seen as a computational bottleneck in ResNet which PiX alleviates.

**Inference Latency.** Latency or per-frame processing rate is crucial from a deployment perspective. Hence, we analyze latency analysis on five representative GPUs, i.e., three desktop-grade GPUs and two low-powered (10W) embedded computing devices far less powerful. See Table A2 in Appendix.

We observe that the impact of PiX is more pronounced on low-powered devices. Particularly on embedded devices, ResNet-50+PiX is 16% faster, ResNet-101+PiX is 14% faster, and ResNet-152+PiX is 15% faster. Considering the extensive usage of low-powered embedded computing devices in real-time applications, the aforementioned improvements are quite advantageous. See the appendix for the latency measures.

## 4.2 PiX as Network Downscalar (Sec. 3.4.2)

Along with Fast Channel Squeezing, PiX also offers simplified network downscaling while achieving better performance than the correspondingly scaled baseline in the same architecture family as referring to Table 3. Note that we used width scaling (increasing the number of channels in each conv layer) for the baseline.

The empirical result in Table 3 shows that our proposed PiX is seamlessly applicable for network downscaling regardless of network architectures (ResNet-18, ResNet-50, VGG-15, and even MobileNet-v1), showing superior performance than all the baselines. It shows the diverse scope and applicability of PiX in low-powered devices for customizing a network for a dedicated purpose.

## 4.3 PiX as Dynamic Channel Pruner (Sec. 3.4.3)

PiX conducts similar behaviors with dynamic pruning while dynamic pruning turns off a few channels (Sec. 3.4.3). We compare PiX with dynamic pruning approaches.

Table 4: PiX as Dynamic Channel Pruner. The experiment is done with ResNet-18

| Approach @ ResNet-18 | Dynamic | Top-1% ↑ | | FLOPs Saving ↑ |
|---|---|---|---|---|
| | | Baseline | Accelerated | |
| ● Soft Filter Pruning He et al. (2018a) | | 70.28 | 67.10 | 1.72× |
| ● Discrimination-aware Channel Pruning Zhuang et al. (2018) | | 69.64 | 67.35 | 1.89× |
| ● Low-cost Collaborative Layers Dong et al. (2017) | ✓ | 69.98 | 67.33 | 1.53× |
| ● Channel Gating Neural Networks Hua et al. (2019) | ✓ | 69.02 | 67.40 | 1.61× |
| ● Feature Boosting and Suppression Gao et al. (2018) | ✓ | 70.71 | 68.17 | 1.98× |
| ● Storage Efficient Pruning Chen et al. (2020) | ✓ | 69.76 | 68.73 | 1.94× |
| ● Manifold Regularized Pruning Tang et al. (2021) | ✓ | 69.76 | 68.88 | 2.06× |
| ● Dynamic Structure Pruning Park et al. (2023) | ✓ | 69.76 | 68.38 | 2.56× |
| ● PiX | ✓ | **73.15** | **70.60** | 1.85× |

Table 5: PiX as Dynamic Channel Pruner. The experiment is done with VGG-16.

| Approach @ VGG-16 | Dynamic | Δ Top-5 ↑ | FLOPs Saving ↑ |
|---|---|---|---|
| ● Filter Pruning Li et al. (2016) | | −8.6 | 4× |
| ● Runtime Neural Pruning Lin et al. (2017) | ✓ | −2.32 | 3× |
| ● AutoML for Model Compression He et al. (2018b) | | −1.4 | 5× |
| ● ThiNet-Conv Luo et al. (2017) | | −0.37 | 3× |
| ● Feature Boosting and Suppression Gao et al. (2018) | ✓ | **−0.04** | 3× |
| ● PiX | ✓ | **−0.04** | 3× |

**ResNet-18.** Referring to Table 4, the PiX baseline (i.e., ResNet-18+PiX @$\zeta = 1$, Table 3) and the downscaled (ResNet-18+PiX @$\zeta = 3$, Table 3), shows better performance than the dynamic pruning approaches. Note that PiX does not require fine-tuning to obtain better performance, unlike other approaches, such as (Gao et al., 2018), leading to a simpler pipeline of PiX.

**VGG-16.** Following (Li et al., 2016; Lin et al., 2017; Gao et al., 2018), we report $\Delta$*Top-5* error with the benefit of FLOP reduction. Table 5 shows that PiX offers a competitive performance.

**Dynamic Channel Pruning as Channel Squeezer.** To further prove the significance of PiX, we conduct a reverse experiment. We customize FBS (Gao et al., 2018) for channel squeezing. FBS picks top-k channels in its original operation and has the same input-output dimensions, i.e., $\in \mathbb{R}^{C \times H \times W}$. Instead, we configure it to output $\in \mathbb{R}^{\lceil C/k \rceil \times H \times W}$, where $k = \zeta$.

By observing FBS, we face the convergence issue. We identify the underlying cause is due to the drop-out of intermediate channels from the input $X$ when selecting top-k channels. Also, the channels appearing in the output ($Y$) that lost position identity or channel index causes convergence issues. When $Y$ is operated upon via subsequent convolutions, the approach is not intended to learn the relation between the channels, as the position or index of a given channel in $X$ keeps changing in $Y$. This indicates that pruning methods can not complement PiX but vice-versa is feasible, highlighting the utility of PiX.

## 4.4 PiX in the Wild

This experiment compares PiX with existing works in improving ResNet accuracy and feature fusion via the attention mechanism. Table 6 shows the analysis.

**E0-E2** We compare PiX with the methods that aim to improve performance with the newly proposed layer, such as (Hu et al., 2018; Woo et al., 2018). We could observe that PiX performs better, even on MobileNet (Howard et al., 2017), while the proposed PiX has a simpler structure (Figure A2) and has multi-purpose utility.

**E3** We also compare PiX with recent Attentional-Feature-Fusion (AFF) (Dai et al., 2021) which fuses two feature maps adaptively, and SKNet (Li et al., 2019) which improves accuracy by adaptively weighting the output of two convolutions with different kernel sizes. These models are trained for longer epochs, therefore we also train PiX at the same setting (Dai et al., 2021). We observe that PiX outperforms these two methods while being architecturally simple and serving other purposes as well.

**E4** RepVGG (Ding et al., 2021) is a recent approach that speeds up VGG (Simonyan & Zisserman, 2014) via structural reparameterization (Sec. 2). We see that VGG-16+PiX offers a competitive performance to RepVGG while being simpler at both train and test time.

Table 6: PiX *vs* Existing approaches. '*' denoting PiX being used only before the second layer in Figure A1c.

| | Approach | #Params ↓ | FLOPs ↓ | Top-1% ↑ |
|---|---|---|---|---|
| E0 | ● ResNet-18 He et al. (2016) + SE Hu et al. (2018) | 11.78M | 1.81B | 70.59 |
| | ● ResNet-18 He et al. (2016) + CBAM Woo et al. (2018) | 11.78M | 1.81B | 70.73 |
| | ● ResNet-18 He et al. (2016) + PiX* | 11.88M | 1.81B | **71.65** |
| | ● ResNet-18 He et al. (2016) + PiX | 12.80M | 1.84B | **73.15** |
| E1 | ● ResNet-50 + SE Hu et al. (2018) | 28.09M | 4.13B | 76.85 |
| | ● ResNet-50 + CBAM Woo et al. (2018) | 28.09M | 4.13B | 77.34 |
| | ● ResNet-50 + PiX | 28.08M | 4.13B | **77.65** |
| E2 | ● MobileNet Howard et al. (2017) + SE Hu et al. (2018) | 5.07M | 0.57B | 70.03 |
| | ● MobileNet Howard et al. (2017) + CBAM Woo et al. (2018) | 5.07M | 0.57B | 70.99 |
| | ● MobileNet Howard et al. (2017) + PiX | **4.06M** | **0.33B** | **72.27** |
| E3 | ● ResNet-50 + AFF Dai et al. (2021)@160 Epochs | 30.30M | 4.30B | 79.10 |
| | ● ResNet-50 + SKNet Li et al. (2019) @160 Epochs | 27.70M | 4.47B | 79.21 |
| | ● ResNet-50 + PiX @160 Epochs | 28.08M | 4.13B | **79.40** |
| E4 | ● RepVGG-A0 Ding et al. (2021) | 9.10M | **1.51B** | 72.41 |
| | ● VGG-16 Simonyan & Zisserman (2014) + PiX | **8.65M** | 1.94B | **72.47** |
| E5 | ● ResNet-50 + DWP Hussain & Hesheng (2019) | 19.60M | 2.82B | 75.35 |
| | ● ResNet-50 + PiX@$\zeta = 2$ | **14.08M** | **2.12B** | **76.65** |

Table 7: PiX *vs* ResNet. Transfer learning evaluation for classification (E0) and semantic segmentation (E1).

| | Architecture | #Param | FLOPs ↓ | CIFAR-10 ↑ | CIFAR-100 ↑ | CityScapes ↑ |
|---|---|---|---|---|---|---|
| E0 | ● ResNet-50 He et al. (2016) | 25.5M | 4.12B | 95.57 | 81.60 | − |
| | ● ResNet-50+PiX | 25.5M | **3.18B** | **95.67** | **82.22** | − |
| E1 | ● Zhao et al. (2017)+ResNet-101 He et al. (2016) | 44.5M | 7.85B | − | − | 78.4 |
| | ● Zhao et al. (2017)+ResNet-101+PiX | 44.5M | **6.05B** | − | − | **79.1** |

**E5** Depth-wise pooling (DWP) (Hussain & Hesheng, 2019) is a comparable approach for channel squeezing. Hence, we trained ResNet-50 endowed with DWP. As mentioned in Sec. 3.4.1, eliminating sampling probability predictor $\phi$ from the network removes all the squeezing layers, leading to parameter and accuracy loss. DWP is an example of this case, which eliminates all the $1 \times 1$ squeezing layers, facing a loss of accuracy (1.30%), compared to PiX used for channel squeezing. Due to the parameter differences in ResNet-50+PiX and ResNet-50+DWP, we compare the latter with a downscaled variant of ResNet-50+PiX. As a result, PiX surpasses DWP, verifying our hypothesis that in channel squeezing mode, PiX preserves the non-linearity that allows for maintaining accuracy.

## 4.5 TRANSFER LEARNING

**E0** To analyze the generalization of PiX across datasets and tasks, we perform transfer learning from ImageNet to CIFAR-10 and CIFAR-100. Each of the datasets consists of 50K training and 10K test images. For training, we finetune the models pretrained over ImageNet. The training strategy for both datasets remains identical to that of ImageNet except for 200 epochs. From Table 7, it can be seen that PiX performs better at lower FLOP requirements.

**E1** We evaluate PiX for a challenging task of semantic segmentation. We use a prominent approach (Zhao et al., 2017) and replace the backbone with ResNet-101+PiX. Consequently, PiX outperforms the baseline both in terms of FLOPs and accuracy by 0.7% units mIoU (mean intersection over union), indicating that PiX transfers well across tasks and datasets.

## 5 CONCLUSION

In this work, we introduce Pick-or-Mix (PiX) for dynamic channel sampling. It works by exploiting global spatial context by blending cross-channel information and then picking or mixing channels on *per-pixel basis*. The picked channels can be different for each pixel depending upon the operator selection. This capability allows PiX to maintain accuracy even by cutting down FLOPs. PiX can work as a computationally efficient channel squeezer, can downscale a given model, or function as a dynamic channel pruner. We show that PiX is easy to plug into the existing ConvNets or even ViT, without altering its structure, and we show that PiX outperforms state-of-the-art approaches.

**Limitations.** Currently, our approach is designed for discrete squeezing factors $\zeta$. Future extensions of the proposed approach include developing a more generalized fusion approach that can sample channels at non-integer $\zeta$.

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

# APPENDIX

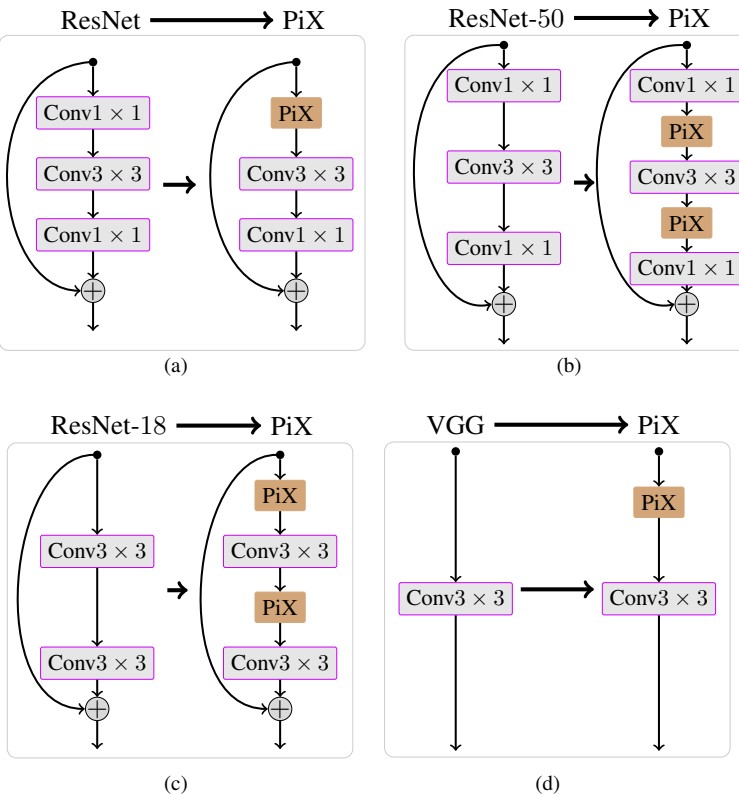

Figure A1: Embedding the proposed PiX into various standard networks for various purposes. (a) **Channel Squeezing Mode**: we *replace* $1 \times 1$ *channel squeezing layers* in ResNet (He et al., 2016) with PiX, where the remaining $1 \times 1$ conv layers in the original ResNet are untouched as it is intended for expanding channel dimensions. (b and c) **Network Downscaling Mode**: We insert PiX modules into ResNet and VGG (Simonyan & Zisserman, 2014). We make the output channel dimension smaller than the input channel dimension by adjusting sampling factor $\zeta$ in PiX. In other words, depending on $\zeta$, The input and output channel dimensions of $1 \times 1$ and $3 \times 3$ conv layers change accordingly. As a result, as $\zeta$ gets larger, the channel dimension of the original network reduces. (c and d) **Dynamic Channel Pruning**: These configurations are used for comparing PiX with other dynamic channel pruning approaches.

## A  PiX INSTANTIATION

Figure A1 shows how one can use PiX in different network architectures and for different tasks.

## B  DIFFERENCE WITH EXISTING MODULES

Figure A2 shows visual differences with the existing modules which aims at accuracy improvement and dynamic pruning approaches.

## C  MOBILENET WITH PiX RESULTS

We also compare PiX embedded into MobileNet-v1 with the approaches of improving network accuracy via attention mechanism. Table A1

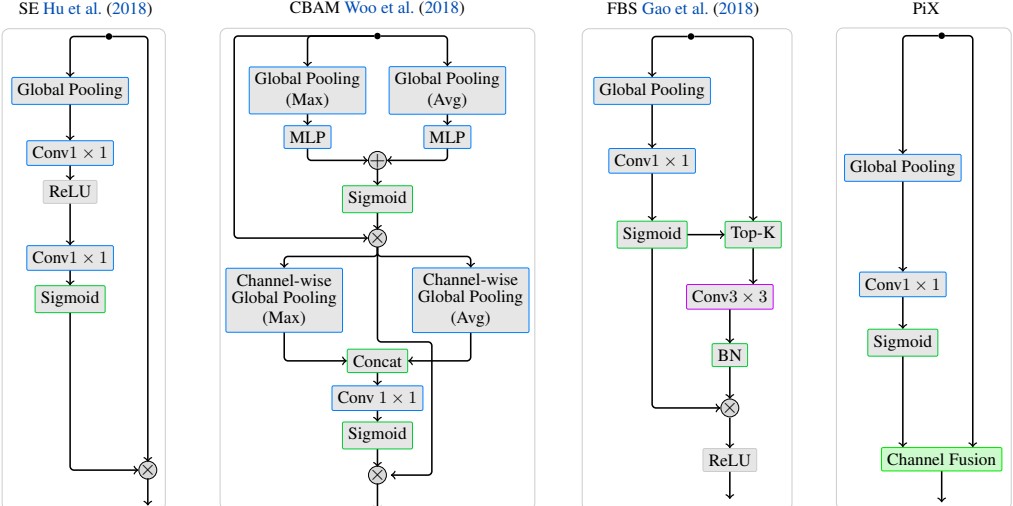

Figure A2: PiX vs existing modules.

Table A1: PiX *vs* Existing attention based approaches for accuracy improvements on MobileNet-v1 Howard et al. (2017). '↑' is better. '↓': is better.

|  | Approach | #Params ↓ | #FLOPs ↓ | Top-1 (%) ↑ |
|---|---|---|---|---|
| E0 | ● MobileNet Howard et al. (2017) + SE Hu et al. (2018) | 5.07M | 0.57B | 70.03 |
|  | ● MobileNet Howard et al. (2017) + CBAM Woo et al. (2018) | 5.07M | 0.57B | 70.99 |
|  | ● MobileNet Howard et al. (2017) + PiX @$\zeta = 2$ | **4.06M** | **0.33B** | **72.27** |

Table A2: Latency analysis of PiX as Channel Squeezer. PiX *vs* ResNet @$224 \times 224$, @FP32, mean of 25 runs.

| Compute Platform NVIDIA | Cores | Compute power | ● Res-50 | ● Res-50 +PiX | ● Res-101 | ● Res-101 +PiX | ● Res-152 | ● Res-152 +PiX |
|---|---|---|---|---|---|---|---|---|
| A40 | 10752 | 37.00 TFLOPs | 7ms | 6ms | 11ms | 10ms | 15ms | 14ms |
| RTX2080Ti | 4352 | 13.45 TFLOPs | 8ms | 6ms | 14ms | 12ms | 17ms | 15ms |
| GTX1080Ti | 3584 | 11.45 TFLOPs | 9ms | 7ms | 13ms | 12ms | 17ms | 15ms |
| Jetson NX | 384 | 1.00 TFLOPs | 48ms | 40ms | 75ms | 64ms | 100ms | 85ms |
| Jetson Nano | 128 | 0.23 TFLOPs | 140ms | 130ms | 230ms | 200ms | 320ms | 280ms |

Table A3: PiX as Channel Squeezer in Recent EfficientViT (Liu et al., 2023) Transformers. The baseline results are reproduced from their official repository `https://github.com/microsoft/Cream/tree/main/EfficientViT` under default training and augmentation hyperparameters as suggested in their repository.

| Architecture | $\zeta$ | #Param | FLOPs ↓ | Top-1% ↑ |
|---|---|---|---|---|
| ● EfficientViT-M5 Liu et al. (2023) | 2 | 12M | 522M | 73.19% |
| ● EfficientViT-M5+PiX | 2 | 12M | **376M (↓27%)** | **73.87 (↑0.68%)** |

## D EFFICIENTVIT WITH PIX RESULTS

We also test our PiX for recent vision Transformers EfficientViT Liu et al. (2023). We replace the channel squeezing layers in all of the FFN of the EfficientViT with out PiX-based channel squeezing. We observe that PiX performs better by $0.68\%$ at $27\%$ fewer FLOPs (Table A3). This indicates that our PiX also applies to the Transformers.

# E    INFERENCE LATENCY

Latency or per-frame processing rate is crucial in practical applications. Hence we show a latency analysis on five representative GPUs (Table A2). The first three are desktop-grade GPUs, while the last two are low-powered (10W) embedded computing devices that are far less powerful.

From the table, the difference in latency can be observed. It is mostly attributed to the variation in the number of computing elements or cores. Theoretically, more cores should run a network faster, however owing to the sequential linking of layers, a layer must wait until the preceding ones finish. This causes similar latency for the first three GPUs, however for them, gains can be examined during batched inference and training, which is a measure of throughput, and is reflected via days-long reduction in training time (Table 2).

In contrast, the impact of PiX is more pronounced on low-powered devices, where the cores are a scarce resource. On Jetson-NX, ResNet-50+PiX is 16% faster, ResNet-101+PiX is 14% faster, and ResNet-152+PiX is 15% faster. On Jetson-Nano, ResNet-50+PiX is 7% faster, ResNet-101+PiX is 13% faster, and ResNet-152+PiX is 12% faster. Notably, these speed-ups can be further enhanced via half-precision (FP16) or Int8 precision, which speed-ups roughly by $2 - 4$ times. Considering the extensive usage of low-powered embedded computing devices in real-time applications, the aforementioned improvements are quite advantageous.

# F    COMPUTATIONAL COMPLEXITY

We show here how PiX achieves computationally efficient channel sampling. However, for better understanding, we first discuss the FLOPs of different kinds of layers.

## F.1    CONVOLUTION

Consider a convolution layer having $N$ kernels and an input feature map $\mathbf{X} \in \mathbb{R}^{C \times H \times W}$. The size of each kernel can be given by $C \times k \times k$. FLOPs for convolution operation are determined using Fusion-Multi-Addition (FMA) instructions. Therefore, the computational demands of a convolution layer can be given as follows:

$$\#\text{FLOPs} = H \times W \times N \times C \times k \times K \tag{4}$$

## F.2    BATCHNORM

The BatchNorm (Ioffe & Szegedy, 2015) operation is performed per spatial location and can be given as $\hat{\mathbf{X}} = (\mathbf{X} - \mu)\frac{\gamma}{\sigma} + \beta$. It can be implemented in three FLOPs, i.e., first for computing $X - \mu$, second for $\gamma/\sigma$, and last as FMA with $\beta$. In general, $\sigma$ is stored as $\sigma^2$, therefore, it requires to compute square-root of $\sigma^2$ to obtain $\sigma$. Overall, it takes four FLOPs to implement a BatchNorm operation per spatial location. Thus, the total number of FLOPs for a BatchNorm layer can be given as:

$$\#\text{FLOPs} = 4 \times C \times H \times W \tag{5}$$

Optionally, during inference, BN can be fused with a Conv operation where convolution is followed by BN, but we remain agnostic to such cases to account for the training phase and other architectures.

## F.3    RELU

A ReLU operation is given by $\mathbf{Y} = X$ for $X \geq 0$ and $\mathbf{Y} = 0$ for $X < 0$. It simply requires a comparison instruction, leading to the total number of FLOPs given by:

$$\#\text{FLOPs} = C \times H \times W \tag{6}$$

## F.4    SIGMOID

A Sigmoid operation is given by $\mathbf{Y} = {1}/{1+\exp^{-\mathbf{x}}}$. It can be implemented in four FLOPs. Therefore, the total FLOPs for a Sigmoid layer can be given by:

$$\#\text{FLOPs} = 4 \times C \times H \times W \tag{7}$$

### F.5 GLOBAL POOLING

Apart from the above layers, in the PiX module, a global pooling operation is also performed. There are several ways to implement a global pooling operation. However, the most common is by using matrix multiplication routines and Fused-Multiply-Add (FMA) instructions. The whole channel of a feature map can be considered as a vector of size $H \times W$ which can be reduced to a scalar by taking its dot product with a vector whose all elements are equal to one. Hence, the total number of FLOPs for the global pooling operation can be given by:

$$\#\text{FLOPs} = C \times H \times W \tag{8}$$

### F.6 CHANNEL SAMPLING

Channel fusion operates on $(C/\zeta)$ subsets, each of $\zeta$ channels. For the `Max` operation, $(\zeta - 1)$ compare instructions, while for `Avg` operation, $(k - 1)$ FMA instructions are required per-location i.e. $\Gamma_{hw}$. Thus, the total number of FLOPs for channel sampling can be given by:

$$\#\text{FLOPs} = (\zeta - 1) \times (C/\zeta) \times H \times W \tag{9}$$

The computational complexity of the PiX block can be calculated based on the several equations developed above.

## G COMPUTATIONS & MEMORY REQUIREMENTS

By using the above equations, we can easily compute the FLOP overhead of various modules such as SE (Hu et al., 2018), CBAM (Woo et al., 2018), or FBS (Gao et al., 2018), demonstrated below how to achieve that:

### G.1 SE (HU ET AL., 2018)

COMPUTE

$$\#\text{Global\_pool\_FLOPs} = C \times H \times W \tag{10}$$
$$\#\text{Conv\_Sqz\_FLOPs} = (C/16) \times C \tag{11}$$
$$\#\text{ReLU\_FLOPs} = (C/16) \tag{12}$$
$$\#\text{Conv\_Exp\_FLOPs} = C \times (C/16) \tag{13}$$
$$\#\text{Sigmoid\_FLOPs} = 4 * C \tag{14}$$
$$\#\text{Broadcast\_Multiply\_FLOPs} = C \times H \times W \tag{15}$$

#Total Flops $= 2CHW + 0.125C^2 + (65/16)C$.

MEMORY

$$\#\text{Global\_pool\_Mem} = C \tag{16}$$
$$\#\text{Conv\_Sqz\_Mem} = C/16 \tag{17}$$
$$\#\text{Conv\_Exp\_Mem} = C \tag{18}$$
$$\#\text{Broadcast\_Multiply\_Mem} = C \times H \times W \tag{19}$$

#Total Memory $= CHW + (33/16)C$.

*Note*: ReLU and Sigmoid are ignored in memory due to their In-place operations.

## G.2 CBAM (WOO ET AL., 2018)

COMPUTE

$$\#\text{Global\_Max\_pool\_FLOPs} = C \times H \times W \tag{20}$$
$$\#\text{Global\_Avg\_pool\_FLOPs} = C \times H \times W \tag{21}$$
$$\#\text{Conv\_Sqz\_FLOPs} = (C/16) \times C \tag{22}$$
$$\#\text{ReLU\_FLOPs} = (C/16) \tag{23}$$
$$\#\text{Conv\_Exp\_FLOPs} = C \times (C/16) \tag{24}$$
$$\#\text{Sigmoid\_FLOPs} = 4 * C \tag{25}$$
$$\#\text{Sum\_FLOPs} = C \tag{26}$$
$$\#\text{Broadcast\_Multiply\_FLOPs} = C \times H \times W \tag{27}$$
$$\#\text{Channel\_Max\_Pool\_FLOPs} = (C-1) \times H \times W \tag{28}$$
$$\#\text{Channels\_Avg\_Pool\_FLOPs} = (C-1) \times H \times W \tag{29}$$
$$\#\text{Concat\_FLOPs} = 2 \times H \times W \tag{30}$$
$$\#\text{Conv\_FLOPs} = 1 \times 2 \times H \times W \tag{31}$$
$$\#\text{Sigmoid\_FLOPs} = 4 \times 1 \times H \times W \tag{32}$$
$$\#\text{Broadcast\_Multiply\_FLOPs} = C \times H \times W \tag{33}$$

$\#\text{Total Flops} = 6CHW + 0.125C^2 + (81/16)C + 6HW.$

MEMORY

$$\#\text{Global\_Max\_pool\_Mem} = C \tag{34}$$
$$\#\text{Global\_Avg\_pool\_Mem} = C \tag{35}$$
$$\#\text{Conv\_Sqz\_Mem} = C/16 \tag{36}$$
$$\#\text{Conv\_Exp\_Mem} = C \tag{37}$$
$$\#\text{Sum\_Mem} = C \tag{38}$$
$$\#\text{Broadcast\_Multiply\_Mem} = C \times H \times W \tag{39}$$
$$\#\text{Channel\_Max\_Pool\_Mem} = H \times W \tag{40}$$
$$\#\text{Channels\_Avg\_Pool\_Mem} = H \times W \tag{41}$$
$$\#\text{Concat\_Mem} = 2 \times H \times W \tag{42}$$
$$\#\text{Conv\_Mem} = H \times W \tag{43}$$
$$\#\text{Broadcast\_Multiply\_Mem} = C \times H \times W \tag{44}$$

$\#\text{Total Memory} = 2CHW + 5HW + (65/16)C.$

## G.3 FBS (GAO ET AL., 2018)

COMPUTE

$$\#\text{Global\_pool\_FLOPs} = C \times H \times W \tag{45}$$
$$\#\text{Conv\_Sqz\_FLOPs} = C \times C \tag{46}$$
$$\#\text{Sigmoid\_FLOPs} = 4 \times C \tag{47}$$
$$\#\text{Top-k\_FLOPs} = \sum_{i \in [1,k]} (C - i) \tag{48}$$
$$\#\text{BatchNorm\_FLOPs} = 4 \times C \times H \times W \tag{49}$$
$$\#\text{Broadcast\_Multiply\_FLOPs} = C \times H \times W \tag{50}$$
$$\#\text{ReLU\_FLOPs} = C \times H \times W \tag{51}$$

$\#\text{Total Flops} = 7CHW + C^2 + 4C + \sum_{i \in [1,k]}(C-i).$

MEMORY

$$\#\text{Global\_pool\_Mem} = C \tag{52}$$

$$\#\text{Conv\_Sqz\_Mem} = C \tag{53}$$

$$\#\text{Top-k\_Mem} = C \times H \times W \tag{54}$$

$$\#\text{Broadcast\_Multiply} = C \times H \times W \tag{55}$$

$\#\text{Total Memory} = 2CHW + 2C$.

*Note*: In memory, BatchNorm is ignored due to its In-place operations.

### G.4   PIX

COMPUTE

$$\#\text{Global\_pool\_FLOPs} = C \times H \times W \tag{56}$$

$$\#\text{Conv\_Sqz\_FLOPs} = (C/\varsigma) \times C \tag{57}$$

$$\#\text{Sigmoid\_FLOPs} = 4 * (C/\varsigma) \tag{58}$$

$$\#\text{Chanl\_Fusion\_FLOPs} = (\varsigma - 1) \times (C/\varsigma) \times H \times W \tag{59}$$

$\#\text{Total Flops} = CHW + C^2 + 4(C/\varsigma) + ((\varsigma - 1)/\varsigma)CHW$.
$\#\text{Total Flops}(@\varsigma = 1) = CHW + C^2 + 4C$.

MEMORY

$$\#\text{Global\_pool\_Mem} = C \tag{60}$$

$$\#\text{Conv\_Sqz\_Mem} = C/\varsigma \tag{61}$$

$$\#\text{Channel Fusion Mem} = C \times H \times W \tag{62}$$

$\#\text{Total Memory} = CHW + ((1 + \varsigma)/\varsigma)C$.

From the above equations, it can be seen that PiX has the lowest FLOPs and lowest Memory required if compared with all the approaches. Values are highlighted in Table A4.

Table A4: This table shows FLOPs and memory usage per-instance of different Modules corresponding to the Figure A3. These values are computed at different height and width of tensor. It can be seen that PiX has lowest FLOP overhead and also requires less memory, equivalent to SE Hu et al. (2018) but half of CBAM Woo et al. (2018) and FBS Gao et al. (2018).

| Method | #FLOPs (M) | #Memory (MB) |
|---|---|---|
| $@\mathbb{R}^{512 \times 112 \times 112}$ | | |
| • SE Hu et al. (2018) | 12.8 | 25.694336 |
| • CBAM Woo et al. (2018) | 38.6 | 51.639424 |
| • FBS Gao et al. (2018) | 45.2 | 51.384320 |
| • PiX | **6.6** | **25.694208** |
| $@\mathbb{R}^{512 \times 56 \times 56}$ | | |
| • SE Hu et al. (2018) | 3.2 | 6.426752 |
| • CBAM Woo et al. (2018) | 9.6 | 12.916096 |
| • FBS Gao et al. (2018) | 11.5 | 12.849152 |
| • PiX | **1.8** | **6.426624** |
| $@\mathbb{R}^{512 \times 28 \times 28}$ | | |
| • SE Hu et al. (2018) | .837 | 1.609856 |
| • CBAM Woo et al. (2018) | 2.4 | 3.235264 |
| • FBS Gao et al. (2018) | 3.0 | 3.215360 |
| • PiX | **0.6** | **1.609728** |

## H   COMPUTATION REDUCTION BY PIX IN CHANNELS SQUEEZING I.E. $\varsigma > 1$

In the baseline method, the squeeze layer operates upon $\mathbf{X} \in \mathbb{R}^{C \times H \times W}$ which requires $C/\varsigma \times C \times H \times W$ FLOPs. Whereas in PiX, the global context aggregation requires $C \times H \times W$ FLOPs, cross-channel

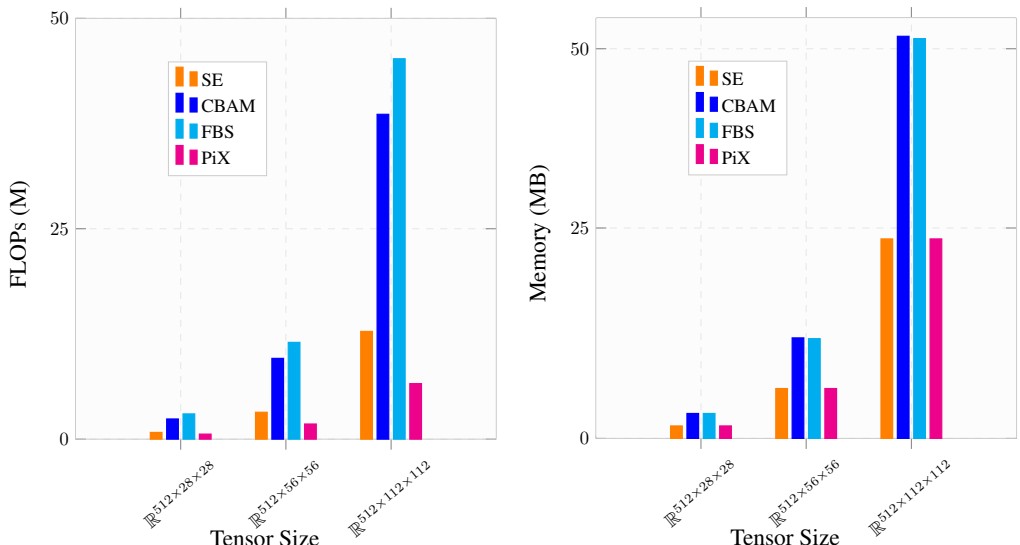

Figure A3: Flops and Memory performance of PiX in contrast to SE Hu et al. (2018) CBAM Woo et al. (2018), and FBS Gao et al. (2018) per-instance of a module. In the memory plot, SE and PiX has almost same overhead but PiX lesser than SE in terms of Bytes ($\sim 1000$), and same is with CBAM and FBS. For this reason plots are overlapping in the memory plot. The actual values are also highlighted in Table A4.

information blending requires $C/\varsigma \times C$ FLOPs. and channel fusion requires $C/\varsigma \times (\zeta - 1) \times H \times W$ FLOPs.

As an example, consider an input tensor $\mathbf{X} \in \mathbb{R}^{12 \times 5 \times 5}$ to a squeeze layer kernels of size $1 \times 1$. With $\zeta = 4$, the number of subsets becomes $12/\zeta = 3$. From the equations discussed, the total number of FLOPs for a squeeze layer equals $1275$.

$$\text{\#Conv\_FLOPs} = 5 \times 5 \times 3 \times 12 \times 1 \times 1 = 900 \tag{63}$$

$$\text{\#BN\_FLOPs} = 4 \times 3 \times 5 \times 5 = 300 \tag{64}$$

$$\text{\#ReLU\_FLOPs} = 3 \times 5 \times 5 = 75 \tag{65}$$

On the other hand, the FLOPs for the PiX module with $\zeta = 4$ equals only $811$, as described below.

$$\text{\#Pooling\_FLOPs} = 12 \times 5 \times 5 = 300 \tag{66}$$

$$\text{\#Conv\_FLOPs} = 1 \times 1 \times 3 \times 12 \times 1 \times 1 = 36 \tag{67}$$

$$\text{\#Sigmoid\_FLOPs} = 4 \times 3 \times 1 \times 1 = 12 \tag{68}$$

$$\text{\#Sampling\_FLOPs} = 3 \times 3 \times 5 \times 5 = 225 \tag{69}$$

In the above example, the baseline squeezing method requires $1275$ FLOPs, whereas PiX requires only $523$ and $748$ FLOPs for PiX and w-PiX fusion strategy respectively. In a similar manner, we achieve huge gains when PiX is plugged into the existing networks, which have been discussed in the experiments section of the paper.

## I  EFFECT OF PICK-OR-MIX ON MEMORY IN CHANNEL SQUEEZING

Despite the computational benefits, PiX does not introduce any memory overhead. The total memory required by the baseline squeeze operation with $\zeta = 4$ can be given by: $\text{\#M} = C/4 \times H \times W$. On the other hand, the memory required for PiX is given by: $\text{\#M} = C + C/4 + C/4 \times H \times W$. We can see that there is a negligible increment in the memory footprint, i.e., from $0.75 \times C \times H \times W$ to $0.75 \times C \times H \times W + 1.25C$. For FP32 precision, the raw memory footprint will be $4 \times M$.

## J  GPU DEPLOYMENT FOR PICK-OR-MIX

The implementation of PiX is quite straightforward and fully parallelizable. For reference, we also have provided the GPU implementation in the supplementary material, naming `pix.cu`. The sam-

pling probability and output feature map computations are parallelizable because they are pointwise operations.

PiX can be implemented directly with the fundamental operators of Pytorch (Paszke et al., 2019). However, since we perform operations over each subset and each location independently, therefore, PiX requires merely $10-15$ lines of NVIDIA's CUDA kernel code or any other parallelization paradigm.

## K   CODES AND IMPLEMENTATION

The code and the pretrained models shall be open-sourced in PyTorch (Paszke et al., 2019). See below for a minimal code snippet.

```python
import torch
import torch.nn as nn
import torch.nn.functional as F
import pix_layer_cuda
import math

# gradients in the backward are received in the order of tensor as they were output in forward function
class PiXOperator(torch.autograd.Function):
    @staticmethod
    def forward(ctx, zeta: int, tau: float,  input: torch.Tensor, fusion_prob: torch.Tensor):
        outputs = pix_layer_cuda.forward(zeta, tau, input, fusion_prob)
        ctx.save_for_backward(input, fusion_prob)
        ctx.zeta = zeta
        ctx.tau = tau
        return outputs[0]

    @staticmethod
    def backward(ctx, out_grad):
        input, fusion_prob = ctx.saved_tensors
        zeta = ctx.zeta
        tau = ctx.tau
        input_grad, fusion_prob_grad = pix_layer_cuda.backward(zeta, tau, input, fusion_prob, out_grad)
        return None, None, input_grad, fusion_prob_grad

class PiXOperatorLayer(torch.nn.Module):
    def __init__(self, zeta, tau = 0.5):
        super(PiXOperatorLayer, self).__init__()
        self.zeta = int(zeta)
        self.tau = tau

    def forward(self, input, fusion_prob):
        return PiXOperator.apply(self.zeta, self.tau, input, fusion_prob)

class PiXLayer(torch.nn.Module):
    def __init__(self, n_ip, zeta, tau=0.5):
        super(PiXLayer, self).__init__()

        n_op = math.ceil(float(n_ip) / zeta)
        self.conv1x1 = torch.nn.Conv2d(n_ip, n_op, 1)

        self.pix = PiXLayer(zeta, tau)
        self.global_pool = torch.nn.AdaptiveAvgPool2d((1, 1))
        self.sigmoid_sqz = torch.nn.Sigmoid()

    def forward(self, x):
        global_pool = self.global_pool(x)
        conv_g_pool = self.conv1x1(global_pool)
        sos_likelihood = self.sigmoid_sqz(conv_g_pool)
        x = self.pix.forward(x, sos_likelihood)
        return x

#### USAGE

n_ip = 24
zeta = 4
pix = PiXLayer(n_ip, zeta)
X = torch.ones([1, n_ip, 4, 4])
Y = pix(X)
print(X)
print(Y)
```

Table A5: Ablation study of ResNet-50+PiX@$\zeta = 4$. Top-1 Accuracy on ImageNet.

| | Ablation | Parameter | Top-1 Accuracy |
|---|---|---|---|
| E0 | Fusion Activation | Sigmoid | 76.77% |
| | | TanH | 76.39% |
| E1 | Batch-Norm | ✗ | 76.77% |
| | | ✓ | 76.44% |
| E2 | $\tau$ | 0.0 | 76.58% |
| | | 0.5 | 76.77% |
| | | 1.0 | 76.54% |
| E3 | Operator | Min | 74.68% |
| | | Max | 76.57% |
| | | Avg | 76.58% |
| | | Max+Avg | 76.77% |

## L    TRAINING SPECIFICATIONS.

The training procedure is kept standard to ensure reproducibility. We use a batch size of 256, which is splitted across 8 GPUs. We use a RandomResized crop (Paszke et al., 2019) of 224×224 pixels, along with horizontal flip. We use SGD with Nesterov momentum of 0.9, base_lr=0.1 with CosineAnnealing (Loshchilov & Hutter, 2016) rate scheduler and a weight decay of 0.0001. Unless otherwise stated, all models are trained from scratch for 120 epochs following (He et al., 2016).

## M    ABLATION STUDY

We empirically validate Pick-or-Mix design practices using the most pertinent ablations possible. ResNet-50 is adopted as the baseline for this purpose, and channel squeezing mode. To begin with, we first analyze the effect of changing the activation function in the cross-channel information blending stage and then examine the effect of placing a BatchNorm prior to the sigmoidal activation. Further, we verify the behavior of proposed channel fusion strategies and also the effect of varying fusion threshold $\tau$.

**E0: Fusion Activation.**    The channel fusion stage utilizes the sampling probability $p$. Given that the value of $p$ lies in the interval $[0, 1]$, we wish to examine the behavior of PiX if this range is achieved via a different activation function. For this purpose, we select TanH function which natively squeezes the input into a range $[-1, 1]$. Therefore, we rewrite the mathematical expression to $0.5 * (1 + \text{TanH})$ in order to place the output of TanH into the desired range of $[0, 1]$. We replace the sigmoidal activation with the above expression and retrain the network. From Table A5, it can be seen that sigmoidal activation is superior to TanH activation for the case of PiX.

**E1: BatchNorm in Global Context Aggregation.**    Out of curiosity, we also analyze the behavior of PiX module by placing a BatchNorm (Ioffe & Szegedy, 2015) after the sampling probability predictor because the squeeze layer in the baseline method is also followed by a BatchNorm layer. We observe that BatchNorm negatively impacts performance.

**E2: Effect of Fusion Threshold ($\tau$).**    The hyperparameter $\tau$ is evaluated against three values $\in \{0.0, 0.5, 1.0\}$. In accordance with the Eq. 2, $\tau = 0$ corresponds to Max operator, $\tau = 1.0$ corresponds to Avg operator regardless of the value of $p$. Whereas $\tau = 0.5$ offers equal opportunity to the Max and Avg fusion operators which are adaptively taken care of by the value of $p$. We present an ablation over the aforementioned three values of $\tau$. From Table A5, it is observable that $\tau = 0.5$ attains the best performance, which is the case when the network has the flexibility to choose between the two reduction operators adaptively. Hence, in the subsequent experiments, we use $\tau = 0.5$ for threshold-based fusion.

**E4: Effect of Operator Type.**    We also experiment for operator Min other than Max and Avg. We found out that Min performs severely worse. This justifies our choice of operators and is in line with the performance achieved by using the pooling operation when they are used spatially.

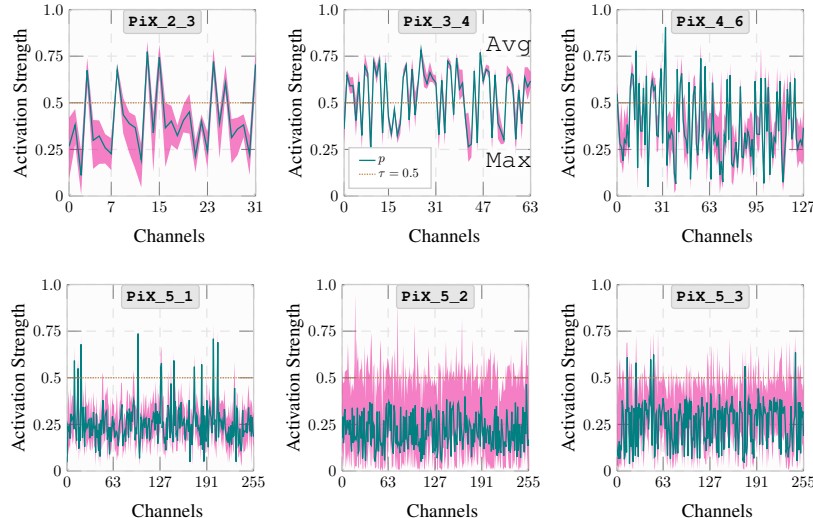

Figure A4: Sampling probability at different stages of ResNet-50+PiX. Stage named as: PiX_STAGE_ID_BLOCK_ID He et al. (2016).

## N  ROLE OF FUSION PROBABILITY

We analyze the sampling probabilities across all classes in the ImageNet validation set for ResNet-50+PiX@PiX,$\zeta = 2$ for the last block of each stage (Figure A4).

It can be seen that importance of probability is significant since distribution for the fusion operator selection is variable i.e. while training, the network does not bias towards only one type of fusion operator, indicating that both of the fusion operators are crucial. In the deeper layers (state-5), the variance starts increasing, indicating deeper layers are class specific and need different activation distribution. This is in line with (Hu et al., 2018). Moreover, we notice that, unlike (Hu et al., 2018), none of the layers in the stage-5 show saturation. This is also an indication that PiX naturally pushes a convolution layer to learn more complex representation.

## O  GRADCAM VISUALIZATION

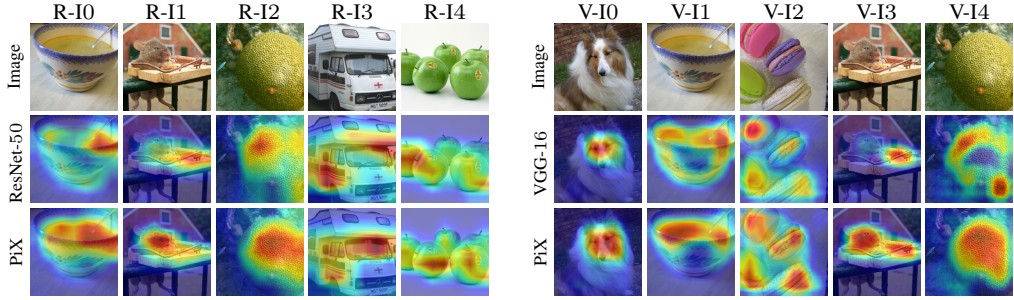

Figure A5: GradCAM for ResNet-50+PiX, VGG-16+PiX. solid red shows more confidence for a pixel to belong to a class.

The performance offered by PiX, especially in the channel squeezing mode inspires us to analyze that how PiX attends the spatial regions relative to the baseline. It explains qualitatively the improved performance of PiX despite the reduction in FLOPs. We use GradCAM (Selvaraju et al., 2017) for this purpose.

Figure A5 shows the analysis for ResNet and VGG. Noticeably, PiX improves the attended regions of a target class relative to the baseline (R-I2, V-I4). Also, in images with multiple instances, PiX focuses on each instance strongly (R-I4, V-I2), indicating that PiX enhances network's generalization by learning to emphasize class-specific parts.

