# OpenReview forum: "Pick-or-Mix: Dynamic Channel Sampling for ConvNets"
_ICLR.cc/2024/Conference — ICLR 2024 Conference Withdrawn Submission_

### Official Review · Reviewer_Nuqz · 2023-10-27

**Soundness:** 2 fair
**Presentation:** 2 fair
**Contribution:** 2 fair
**Rating:** 5
**Confidence:** 5

**Summary:**

This paper proposes a Pick-or-Mix (PiX) module for dynamically sampling channel of ConvNets. Specifically, PiX module introduces a sampling probability predictor to generate weights and select max/average operator, while the selected operators and weights are used to aggregate the features. The introduced PiX module could be adopted in various tasks, such as channel squeezing, network downscaling and dynamic channel pruning. The experiments are conducted on several vision tasks.

**Strengths:**

+: Experimental results show that the proposed PiX module brings performance improvement for backbone models in terms of both accuracy and computational efficiency, while the proposed PiX module could be well generalized to various tasks.

+: The proposed method seems simple and easy to implement.

**Weaknesses:**

-: I have a bit doubt on soundness of the proposed method. Specifically, why the features in the same group can use the same max/average operator? In other words, could the channel sampling probability $p$ for $i$-th element of $z = gca(X)$ represent all channels in $i$-th group? Should all channels in $i$-th group use the same max/average operator? Additionally, I am confused about how different pixels in the same channel adopt different operators.

-: The experiments show the proposed PiX module can bring accuracy improvement over $1 \times 1$ convolution. However, could the authors provide more rigorous theoretical analysis on this phenomenon?

-: Could $\phi(z)$ be implemented by some efficient ways, such as 1D convolution with the stride? Besides, more efficient channel attention or parameter-free attention could be compared to verify the effectiveness of PiX module.

**Questions:**

Other comments:

-: The description on the proposed method is a little confusing, and I suggest that the authors would better give a detailed description on algorithm. The caption of Fig. 2 is too simple, where all symbols lack the explanations and make the readers hard to follow.

-: Could the proposed PiX module be adopted to FNN of ViT?

-: More experiments (e.g., object detection on MS COCO) could be conducted to further verify the effectiveness of PiX module.

---

### Official Review · Reviewer_35yc · 2023-10-30

**Soundness:** 2 fair
**Presentation:** 3 good
**Contribution:** 2 fair
**Rating:** 5
**Confidence:** 5

**Summary:**

This paper proposes a Pick-or-Mix (PiX) module for convolutional neural networks.

The PiX module consists of
(1). Global pooling.
(2). Linear projection to reduce dimension and obtain the sampling probability with Sigmoid activation.
(3) According to the sampling probability, choose whether to do avrage pooling or max pooling for each pixel within a group channels.

Experiments on ImageNet show better speed and performance when compared with ResNet.

**Strengths:**

1. The paper writes clearly and is easy to follow.
2. The proposed PiX module is flexible. It can be used to downscaling network and dynamic channel pruning.

**Weaknesses:**

1. There is no new basic operations in the proposed module.
    SENet by Hu et al. uses global pooling to get the dynamic weights for each channel. CBAM by Woo et al. generates the channel weights considering both max pooling and avg pooling.
    It seems that the proposed module is a combination of SENet, CBAM, and group convolution.
    Morevoer, the improvements compared with SKNet and RepVGG in table 6 are limited.

2. The proposed module also has relations with group convolution.
    However, there is no comparions with it.
    The computational cost could be greatly reduced with group convolution.

3. Is the proposed PiX module applicable to the latest convolutional networks, like ConvNeXt, and ResNeSt?

**Questions:**

See weaknesses.

**Details Of Ethics Concerns:**

No ethics concerns.

---

### Official Review · Reviewer_9dqH · 2023-10-31

**Soundness:** 2 fair
**Presentation:** 4 excellent
**Contribution:** 2 fair
**Rating:** 5
**Confidence:** 4

**Summary:**

The authors introduce a new module for dynamic channel sampling called pick-or-mix (Pix) to replace 1×1 channel squeezing layers in Convnets. Pix first divides a set of channels into subsets and then outputs one channel from each subset via either a max- or a average-pooling operation. The decision to choose between max or mean is determined via an input dependent gating operation. The authors evaluate their method on the ImageNet, CIFAR-10 and CIFAR-100 datasets, and report preliminary results on the CityScapes dataset for semantic segmentation.

**Strengths:**

The paper is very well written and easy to follow. The authors have done a very good job in analyzing the computational cost, memory footprint, and run-time of their proposed solution. The proposed method has been applied to a large number of Convnet architectures and the author's report results on EfficientViT in the appendix as well. The evaluation on 4 datasets is thorough and sufficient.

**Weaknesses:**

I have several concerns about the evaluation and novelty of the proposed method.

- The major component of the proposed method, namely Depth-wise pooling operation has already been proposed in [1] and [2]. The main differentiating factor seems to be the pick operator which learns to dynamically select between Average- and Max-pooling. However, the ablation study presented in the appendix (Table A5) shows that there is no significant increase in performance for having both operators and choosing among them. Top-1 Acc for "only Average pooling" is 76.58% vs 76.77% for Max+average pooling. If the selection between the two operators is the only novelty, the authors should show the added benefit for that. This ablation study should be in the main body and ideally presented for more models considered in the paper.

- I have major concerns regarding fairness of the comparisons. The accuracies for competing methods are copy-pasted from their respective papers with possibly different training pipelines and hyperparameters. Most of the reported baseline numbers for the models in this paper are higher than the ones reported in the reference papers. A major concern is that the gap between the performances of different channel squeezing methods may come from improved baseline accuracies and not the proposed method. Since the accuracies of other methods are copy-pasted, I find the numbers reported in Table 6 and A1 in the supplementary misleading.

- In Table 4, considering the relative drop in performance for different channel pruning methods  with respect to their corresponding baseline accuracies, it seems PiX is performing worse than several methods.



[1] Abid Hussain et al. Depth-Wise Pooling: A Parameter-Less Solution for Channel Reduction of Feature-Map in Convolutional Neural Network

[2] Zhanyu Ma et al. Channel Max Pooling Layer for Fine-Grained Vehicle Classification

**Questions:**

The authors mention PiX does not require fine-tuning to obtain better performance compared to other approaches. Doesn't Pix require learning/fine-tuning $\theta$ and $\beta$ parameters for the samplers?

---

### Official Review · Reviewer_45BR · 2023-11-01

**Soundness:** 3 good
**Presentation:** 3 good
**Contribution:** 2 fair
**Rating:** 3
**Confidence:** 4

**Summary:**

The paper proposes a new method for compressing convolutional neural networks (CNNs). The authors suggest that channel compression is an effective method of compressing CNNs. Based on this idea, they propose a new channel compression scheme. Specifically, the convolutional layers are evenly divided into several parts, and each part is compressed using a certain strategy. The authors propose a dynamic compression scheme that dynamically performs compression based on different inputs. According to their experiments, this approach can effectively compress convolutional neural network structures such as ResNet.

**Strengths:**

1. The problem addressed in this paper is important, and the compression of convolutional neural networks is indeed a worthwhile research topic. The rationale behind this paper is also quite reasonable.
2. The writing in this article is clear, making it easy to understand the introduction of the methods and the description of the experiments.

**Weaknesses:**

1. The experiments are not comprehensive enough as the author only conducted experiments on the ResNet and VGG series network structures. However, there have been many recent advancements in network structures, such as the EfficientNet series or the ViT series. Conducting experiments on a wider range of network structures can enhance the impact of this paper.

2. The comparisons are not sufficient, as many state-of-the-art pruning methods have not been compared. Moreover, compared to other solutions, the improvements presented in this paper are not significant. In Table 7, the method proposed in this paper does not show any advantages over RepVGG.

3. The lack of speed comparisons is a drawback. When it comes to compressing convolutional neural networks, theoretical comparisons based on FLOPs alone are not sufficient. Comparisons based on actual running speeds are more important. The author only compared the speed with the baseline in the appendix and did not compare it with the speeds of other approaches. Furthermore, the speed improvement of the method proposed in this paper is not significant compared to the baseline.

**Questions:**

As shown in the weakness.